# Digital health app development standards: a systematic review protocol

Michelle Helena Van Velthoven,[1] James Smith,[2,3] Glenn Wells,[4] David Brindley[1]

[1]Department of Paediatrics, Healthcare Translation Research Group, University of Oxford, Oxford, UK
[2]The Oxford–UCL Centre for the Advancement of Sustainable Medical Innovation (CASMI) Oxford, University of Oxford, Oxford, UK
[3]Nuffield Department of Orthopaedics, Rheumatology and Musculoskeletal Sciences, University of Oxford, Oxford, UK
[4]Oxford Academic Health Science Centre, Oxford, UK

**Correspondence to**
Dr Michelle Helena Van Velthoven;
michelle.vanvelthoven@paediatrics.ox.ac.uk

## ABSTRACT

**Introduction** There is currently a lack of clear and accepted standards for the development (planning, requirement analysis and research, design and application testing) of apps for medical and healthcare use which poses different risks to developers, providers, patients and the public. The aim of this work is to provide an overview of the current standards, frameworks, best practices and guidelines for the development of digital health apps. This review is a critical 'stepping stone' for further work on producing appropriate standards that can help mitigate risks (eg, clinical, privacy and economic risks).

**Methods and analysis** A systematic review identifying criteria from applicable standards, guidelines, frameworks and best practices for the development of health apps. We will draw from standards for software for medical devices, clinical information systems and medicine because of their relatedness and hope to apply lessons learnt to apps. We will exclude other types of publications, and those published in languages other than English. We will search websites of relevant regulatory and professional organisations. For health apps, we will also search electronic research databases (eg, MEDLINE, Embase, SCOPUS, ProQuest Technology Collection and Engineering Index) because relevant publications may not be found on other websites. We will hand-search reference lists of included publications. The review will focus on international, USA, European and UK standards because these are the markets of primary interest to the majority of app developers currently. We will provide a narrative overview of findings and tabular summaries of extracted data. Also, we will examine the relationship between different standards and compare USA and European Union standards.

**Ethics and dissemination** No ethics approval is required. The review will be disseminated through peer-reviewed publications, conference presentations and inform efforts that aim to improve the quality of health apps through existing links with relevant organisations.

## INTRODUCTION
### Description of the issue

There is a lot of 'apptimism' for the potential of health apps to improve the quality of care and reduce costs.[1] However, despite a rapid growth of the health apps market with an estimated 325 000 health apps available in 2017,[2] this potential has not been achieved. Health apps are software programs that are used in the context of healthcare on mobile

communication devices, such as smartphones and tablets, that can also be used as accessories, such as wearable devices, or as a combination of accessories and software.[3] However, there are many low-quality and unsafe health apps and even apps with potentially harmful content.[4] This situation is resulting in different types of risks for users such as embarrassment, stigma, discrimination, stress, dissatisfaction, delay in effective treatment, poor lifestyle choices and deterioration in health.[5] Also, providers can be negatively impacted by reputation loss, poor quality of care, increase in undue demand on services and opportunity losses.[5]

One of the reasons for the large number of low-quality health apps is that there are no agreed standards for their development, assessment and appraisal. Health apps can be developed quickly, at any place and time by anyone interested, including people with non-medical backgrounds, which can create conflicting views on rapid technology development versus thorough evidence-based medicine principles.[6] Apps are often developed by start-ups with limited resources for research and development which may result in short duration pilots with small participant numbers. Traditional healthcare companies with larger financial resources, such as pharmaceutical companies, on the other hand,

have realised that they need to engage with digital health but are struggling given the differences between the development of drugs and digital tools.[7] As a result, there is a lack of consistency in the development of health apps.

## Description of standards

A standard can be defined as 'a document that provides requirements, specifications, guidelines or characteristics that can be used consistently to ensure that materials, products, processes and services are fit for their purpose'.[8] Standards are collaborative efforts, written by committees of manufacturers, users, research organisations, government departments and consumers.

Medical devices, clinical software and medicines have many standards, regulations and guidances for their development.[9] For example, the International Organization for Standardization (ISO) has a standard on software for medical devices, IEC 62304:2006 'Medical device software—Software life cycle processes' which complements the main standard for medical devices, ISO 13485:2016 'Medical devices—Quality management systems—Requirements for regulatory purposes' and ISO 14971:2007 'Application of risk management to medical devices'.[10] Similarly, in pharmaceutical manufacturing, standards exist such as International Society for Pharmaceutical Engineering (ISPE) Good Automated Manufacturing Practice for computerised systems[11] which are widely adopted.

However, for health software development, there is the concern that standards will inhibit innovation. There needs to be a balance between basic principles for safe and efficient development of health apps that allows products to be built correctly and efficiently. Efforts have been made to develop more proportionate and adaptive governance of innovative technologies for different types of innovation, in different industry sectors.[12]

## The benefits of standards for health apps

Standards can mitigate the risks of health apps, including clinical, privacy and economic risks, which are influenced by the function(s) of the health app, user and contextual factors.[5] Health apps are clinical software and can be divided into higher-risk apps classified as medical devices, such as clinical-decision-support apps, and lower-risk apps that are not, such as wellness and fitness apps.

Standards can help with developing appropriate products that are fit for purpose. Standards can have economic benefits such as contributing to the growth of economies, productivity and Gross Domestic Product (GDP), and exports.[13] For companies, using standards can also enhance their reputation, improve compliance with regulations and encourage innovation through the diffusion of knowledge. For users, standards can ensure the safety, quality and consistency of products.[13]

## Why it is important to do this review

Previous efforts have developed standards for certain health apps, such as the British Standards Institute (BSI)

PASS 277:2015, a standard for quality criteria for health and wellness apps across the life cycle[14] which builds on more established approaches for clinical software such as the Association for the Advancement of Medical Instrumentation TIR45:2012 guidance on the use of agile practices in the development of medical device software.[15] However, such guidance is focused specifically on the UK, and there is a clear need to provide an overview of standards applicable to all health apps across broader jurisdictions. Additionally, understanding and collating the requirements for software development in closely related fields would be useful in informing development of standards at a later date. We will conduct a systematic review to address these needs.

## OBJECTIVES

This systematic review is part of a larger project that addresses the current lack of clear standards for apps for medical and healthcare use and the risk that not having these standards poses to developers, providers, patients and the public. The objectives of this systematic review are to:

1. Provide an overview of currently applicable standards, guidelines, frameworks and best practices relevant for the development of digital health apps.
2. Look at other not directly applicable but related standards to see if relevant lessons can be learnt from current software-specific guidance for medical devices, medication and clinical information systems.

The review will inform efforts that aim to improve the quality of health apps and is a critical 'stepping stone' to further research on producing actionable guidelines for developers and adopters.

## METHODS AND ANALYSIS

This is the protocol for a systematic review that is reported, where possible, according to the Preferred Reporting Items for Systematic Reviews and Meta-Analyses for Protocols[16] which is provided as a supporting document.

### Patient and public involvement

Patients and the public were not involved in writing this protocol.

### Criteria for considering publications

We will include applicable standards, guidelines, frameworks and best practices for the development (planning, requirement analysis and research, design and application testing[14]) of health apps. We will draw from software standards for medical devices, clinical information systems and medicine because of their relatedness and hope to apply lessons learnt to apps.

Standards are requirements, specifications, guidelines or characteristics that can be used consistently to ensure that materials, products, processes and services are fit for their purpose. Guidelines are advice or information

aimed at resolving a problem or difficulty while frameworks are underlying structures for describing a process. A framework is 'a platform for developing software applications. It provides a foundation on which software developers can build programs for a specific platform'.[17] Best practice is a method or technique that has been generally accepted as superior to any alternatives.

An app is defined similarly by different organisations,[3 18] for example, by the US Food and Drug Administration (FDA) as 'software programs that run on smartphones and other mobile communication devices. They can also be accessories that attach to a smartphone or other mobile communication devices, or a combination of accessories and software'.[3] In the context of healthcare, the FDA defines mobile medical apps as 'medical devices that are mobile apps, meet the definition of a medical device and are an accessory to a regulated medical device or transform a mobile platform into a regulated medical device'.[3] The Medicines and Health Regulatory Authority (MHRA) broadly considers health apps to be medical devices if they have a medical purpose (eg, prevention, diagnosis, monitoring, treatment of disease, diagnosis of disease, injury or handicap, compensation for injury or handicap, investigation, replacement of modification of the anatomy or of a physiological process, control of conception).[18] The BSI considers a health or wellness app when it 'contributes to any aspect of the physical, mental or social wellbeing of the user or any other subject of care or wellbeing'.[14]

We will exclude other types of papers, such as editorials, opinion pieces, viewpoints and publications in languages other than English. It will not be possible to provide an overview of standards in all countries around the world given our limited resources. Therefore, we will focus on international, US, European and UK standards because these are the markets of primary interest to the majority of app developers currently.

### Information sources

We will search the following standards databases for health apps, medical devices, clinical software and medicines advised by Imperial College London librarians (2007 until date of search)[19]:

► ISO (https://www.iso.org/obp/ui/#search).
► American National Standards Institute (https://www.ansi.org/).
► European Committee for Standardisation (CEN; https://www.cen.eu/Pages/default.aspx).
► BSI (https://www.bsigroup.com/en-GB/).
► TechStreet (http://www.techstreet.com/).
► IEEE Xplore Digital Library (http://ieeexplore.ieee.org/Xplore/guesthome.jsp).

Furthermore, we will search databases from regulatory and professional organisations for standards on health apps, medical devices, clinical information systems and medicines (2007 until date of search):

► US FDA databases (https://www.fda.gov/default.htm).

► European Medicines Agency (http://www.ema.europa.eu/ema/).
► European Commission (https://ec.europa.eu/info/index_en).
► UK MHRA (https://www.gov.uk/government/organisations/medicines-and-healthcare-products-regulatory-agency).
► The International Council for Harmonisation of Technical Requirements for Pharmaceuticals for Human Use (http://www.ich.org/).
► ISPE (https://www.ispe.org/).
► Advanced Safety in Health Technology (http://www.aami.org/).
► UK National Health Service (NHS) Digital (http://content.digital.nhs.uk/isce/publication/standards).
► Apple app store (https://developer.apple.com/app-store/guidelines/).
► Android app store (https://developer.android.com/distribute/best-practices/launch/launch-checklist.html).

Additionally, relevant articles on guidance, frameworks and best practices for the development of health apps will be identified by searching the following electronic databases (2007 until date of search):

► MEDLINE through Ovid.
► Embase through Ovid.
► Scopus.

### Search strategy

Preliminary draft search strategies for a regulatory website and MEDLINE can be found in the online supplementary file and will be further developed and tailored to the different databases. We will use the titles, abstracts and keywords of a set of articles for which we know that meet our inclusion criteria to define a search strategy that will return all these articles without an unmanageably large number of irrelevant articles. Also, we will hand-search reference lists and ask experts in the field to identify relevant standards.

## STUDY RECORDS
### Selection of studies

All search results will be imported into Zotero reference management software. We will exclude duplicate references by comparing titles, authors and digital object identifiers between similar search results. One reviewer will screen all titles and abstracts of search results independently against the inclusion and exclusion criteria. The second reviewer will screen 10% of these citations to validate the screening process. In case of high disagreement (>10%), the second reviewer will screen all citations. In case of multiple versions of a document, the most recent and most broadly applicable geographically will be selected (ie, the ISO international standard rather than the CEN European standard). One reviewer will retrieve full-text papers. When a full-text paper cannot be obtained, the authors will be contacted with a request

to provide the publication. If no response is received, up to two attempts to contact the authors will be made. Two reviewers will assess full text for eligibility, with any disagreement to be resolved through discussion with a third author. Selection of studies will be reported in a flow chart.

## Data extraction and management

To extract data from included papers, one reviewer will use a standardised Excel form to extract data from included publications (see draft data extraction sheet in the online supplementary file). A second reviewer will validate data extraction by comparing the data extraction sheet with the original publication.

## Data items

The data extraction form will be based on the Reporting Tool for Practice Guidelines in Health Care (the RIGHT Statement[20]) and include basic information (eg, title, year published, focus), background (eg, problem, aim, end-users), evidence (questions, use of systematic reviews), recommendations/requirements (eg, rationale), review and quality assurance, funding, declaration and management of interest and other information (see online supplementary file). The criteria have been adapted to make them relevant to health app development. Quality appraisal will be undertaken by assessing the proportion of items in the adapted RIGHT Statement[20] that are reported in the standards, guidelines, frameworks and best practices.

## Outcomes and prioritisation

The primary outcome is to evaluate and determine the current standards for health app development. Secondary outcomes are to: (1) compare US and EU standards, (2) identify potential limitations in standards based on other software-specific standards, (3) find opportunities to improve existing standards (eg, patient safety, support innovation) and (4) determine and prioritise app development areas for focus in standards development.

## Data synthesis

We will provide a narrative overview of findings and tabular summaries of extracted data. Also, we will analyse the relationship between different standards. Quantitative synthesis is inappropriate for the outcomes of this systematic review. This means that also no assessment of meta-biases and strength of the body of evidence will be undertaken.

## Ethics and dissemination

This review will systematically identify and assess standards, guidelines, frameworks and best practices relevant for the development of health apps. The full systematic review will be submitted for publication in a peer-reviewed medical journal. A possible limitation of this review is that it only focuses on standards reported by international, US, European and UK organisations; however, these are the markets of primary interest to the majority of app developers currently. The review will inform efforts that aim to improve the quality of health apps disseminated through existing links with relevant organisations, such as the BSI, Academic Health Sciences Network, NHS Digital, National Institute for Health and Care Excellence, MHRA, Digital Health and Care Alliance, Digital Health Oxford and London, and US FDA. This evaluation is a critical 'stepping stone' for future work to producing actionable guidelines for developers and adopters.

**Correction notice** Since this paper was first published online the open access licence has changed from CC-BY-NC to CC-BY.

**Contributors** MHVV wrote the protocol. JS, DB and GW provided substantial comments for important intellectual input on the protocol.

**Funding** MHV is a Sir David Cooksey Fellow in Healthcare Translation at the University of Oxford and received no specific additional funding for this work. JS is supported by a UK Medical Research Council Studentship.

**Disclaimer** The funder had no role in writing this protocol.

**Competing interests** The author(s) declared the following potential competing interests with respect to the research, authorship, and/or publication of this article: This article represents the authors' individual opinions and may not necessarily represent the viewpoints of their employers. MHV is the director of Dutches Consulting Ltd which provides digital health-related advice to clients in the life sciences. DB is a stockholder in Translation Ventures Ltd (Charlbury, Oxfordshire, UK) and IP Asset Ventures Ltd (Oxford, Oxfordshire, UK), companies that, among other services, provide cell therapy biomanufacturing, regulatory and financial advice to pharmaceutical clients. DB is also subject to the CFA Institute's codes, standards and guidelines, so he must stress that this piece is provided for academic interest only and must not be construed in any way as an investment recommendation. Additionally, at the time of publication, DB and the organisations with which he is affiliated may or may not have agreed and/or have pending funding commitments from the organisations named here.

**Patient consent** Not required.

**Provenance and peer review** Not commissioned; externally peer reviewed.

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
