## [Reviewer comments · BMJ Open]

ARTICLE DETAILS

TITLE (PROVISIONAL)	Digital health app development standards: a systematic review protocol
AUTHORS	Van Velthoven, Michelle; Smith, James; Wells, Glenn; Brindley, David

VERSION 1 – REVIEW

REVIEWER	Paulina Bondaronek University College London, UK
REVIEW RETURNED	18-Jun-2018

GENERAL COMMENTS	Dear Authors, This is a very timely study and also an ambitious one! Here are just some minor comments regarding the Protocol: First, In Abstract, the authors write: “Also, we will analyze the relationship between different standards and compare US and EU standards.” As no statistical analysis will be used, consider rephrasing. Second, “The aim of this work is to provide an overview of the current standards, frameworks, best practices and guidelines for the development of digital health apps. This review is a critical 'stepping stone' for further work on producing appropriate standards that can help mitigate risks” – by “mitigating risks” does the author mean that they are focusing of safety of the apps? If safety is not the only concern (which seems to be the case as the study focuses on quality) then I 'd suggest to make the aim more specific. Specify what risks. Re:8 References need to be edited, e.g., number 7
--

REVIEWER	Urs-Vito Albrecht Peter L. Reichertz Institute for Medical Informatics, University of Braunschweig – Institute of Technology and Hannover Medical School, Germany
REVIEW RETURNED	19-Jun-2018

GENERAL COMMENTS	No comments
-------------

VERSION 1 – AUTHOR RESPONSE

Editorial requests, reviewers' comments and authors' responses

Requests and comments	Responses
-----------

Editorial Requests

Has this study been registered in the PROSPERO database (<https://www.crd.york.ac.uk/prospero/>)? If so then please include the registration details at the end of the abstract.

No, this study has not been registered in the PROSPERO database because it does not fit their inclusion criteria (e.g. Reviews of methodological issues need to contain at least one outcome of direct patient or clinical relevance in order to be included in PROSPERO).

Please provide the dates of coverage for each database searched (in the methods section)

We have added dates of coverage for each database searched in the methods section.

Reviewer: 1

This is a very timely study and also an ambitious one! Here are just some minor comments regarding the Protocol:

We would like to thank the reviewer for their review and comments.

First, In Abstract, the authors write: “Also, we will analyze the relationship between different standards and compare US and EU standards.” As no statistical analysis will be used, consider rephrasing.

We rephrased this sentence as following: ‘*Also, we will **examine** the relationship between different standards and compare US and EU standards.*’

Second, “The aim of this work is to provide an overview of the current standards, frameworks, best practices and guidelines for the development of digital health apps. This review is a critical ‘stepping stone’ for further work on producing appropriate standards that can help mitigate risks” – by “mitigating risks” does the author mean that they are focusing of safety of the apps? If safety is not the only concern (which seems to be the case as the study focuses on quality) then I ‘d suggest to make the aim more specific. Specify what risks.

Indeed, safety is not only the concern of this systematic review. We rephrased this sentence as following: ‘*This review is a critical ‘stepping stone’ for further work on producing appropriate standards that can help mitigate risks (e.g. **clinical, privacy and economic risks**).*’

Re:8 References need to be edited, e.g., number 7

We have edited the references of the paper, including number 7.

Reviewer: 2

No comments

We would like to thank the reviewer for their review.